# A lagrange programming neural network approach for nuclear norm optimization

**Xiangguang Dai[1]\*, Jian Qiu[2], Chaoyang Wan[2], Facheng Dai[3]**

**1** Chongqing Engineering Research Center of Internet of Things and Intelligent Control Technology, Chongqing Three Gorges University, Chongqing, China, **2** Chongqing Three Gorges University, Chongqing, China, **3** Chongqing University of Posts and Telecommunications, Chongqing, China

\* daixiangguang@163.com

**Data Availability Statement:** All relevant data are within the paper.

**Funding:** The work described in this paper was supported by the Science and Technology Research Program of Chongqing Municipal

## Abstract

This article proposes a continuous-time optimization approch instead of tranditional optimiztion methods to address the nuclear norm minimization (NNM) problem. Refomulating the NNM into a matrix form, we propose a Lagrangian programming neural network (LPNN) to solve the NNM. Moreover, the convergence condtions of LPNN are presented by the Lyapunov method. Convergence experiments are presented to demonstrate the convergence of LPNN. Compared with tranditional algorithms of NNM, the proposed algorithm outperforms in terms of image recovery.

## 1 Introduction

Recently, many researchers were increasingly interested in collecting useful messages from very limited information. Most real data matrices restoring information are low-rank or approximate low-rank. In other words, a fundamental assumption of the problem is that the target matrix has a low-rank structure. Thus, we call similar issues of a low-rank structure to be the Rank Minimization Problem (RMP). Some applications (e.g. control [1], machine learning [2], and computer vision [3]) can be formulated as different versions of RMPs. In the view of applications, RMP can be commonly reformulated as a problem of reconstructing data matrix from partial matrix sampling. Suppose that $M \in \mathbb{R}^{n1 \times n2}$ is a target matrix with partial observations. $\Omega$ is a set of known elements in matrix $M$. $X$ is a low-rank matrix after recovered and $rank(X)$ is the rank of $X$. RMP can be formulated into the following optimization problem

$$
\begin{aligned}
\min \quad & rank(X) \\
s.t. \quad & X_{ij} = M_{ij} \quad (i,j) \in \Omega.
\end{aligned}
$$

(1)

Due to the nonconvexity and combinatorial nature properties of rank functions, problem (1) is an NP-hard problem. Candès et al. [4] proved that the majority missing elements of a matrix with the rank $r$ can be properly restored with a high probability by using a simple convex optimization programme in case of

$$
\begin{aligned}
n &= max(n1, n2), \\
m &\geq Cn^{1.2} r \log n,
\end{aligned}
$$

(2)

Education Commission (Grant No. KJZD-M202201204, KJZD-K202201205), the Opening fund of Chongqing Engineering Research Center of Internet of Things and Intelligent Control Technology (Grant No. zhlv-20221007, zhlv-20221002), Science and Technology Innovation Smart Agriculture Project of Science and Technology Department, Wanzhou District of Chongqing (Grant No. 2022-17), and the Opening Project of Sichuan Province University Key Laboratory of Bridge Non-destruction Detecting and Engineering Computing (Grant No. 2022QYY04). All grants were received by XD.

**Competing interests:** The authors have declared that no competing interests exist.

where $C$ is a certain positive numerical constant. Commonly, the above-mentioned problem can be reformulated as a nuclear norm minimization problem (NNM). The task of fully recovering the majority missing elements $M$ with rank $r$ can be accomplished by NNM. Problem (1) can be rewritten as

$$
\begin{aligned}
\min \quad & \|X\|_* \\
s.t. \quad & X_{ij} = M_{ij} \quad (i,j) \in \Omega,
\end{aligned}
\tag{3}
$$

where $\|X\|_* = \sum_{i=1}^{k} \sigma_i(X)$ is the nuclear norm of $X$, $\sigma(X)$ is the singular value of the matrix $X$ and $k = min(n1, n2)$.

Many optimization approaches [5–12] were proposed to solve problem (3). Cai et al. [7] proposed singular value thresholding (SVT) to address the dual of a regularized approximation of (3) by the linearized Bremgan iterations [11]. Toh et al. [12] proposed an accelerated proximal gradient with linesearch algorithm (APGL) by [6]. Ma et al. [9] proposed fixed point continuation with approximate (FPCA) singular value decomposition (SVD) using a fast Monte Carlo algorithm for SVD calculations. In addition, there are other approaches [5, 8, 10]. Notably, above referenced approaches can generate continuous real-time solutions. Much more room is needed to develop new optimization algorithms of problem (3).

Neural networks were used to address nonlinear equality constraint problems. Many researchers aimed to reformulate the optimization problem (3) and solve it by neural networks [13, 14]. The main advantage of neural networks is to generate real-time solutions. Zhang and Constantinides [15] firstly created a Lagrange programming neural network (LPNN) to address a variety of nonlinear constrained optimization problems. After that, the global convergence conditions of LPNN were attracted the attention [16, 17]. Recently, a large number of variants of LPNNs were proposed to address different application problems. Feng et al. [18] proposed a new LPNN approach to recovere sparse signals based on the locally competitive algorithm (LCA). Liang et al. [19] and Shi et al. [20] applied LPNN to address the radar location problem. Xiong et al. [21, 22] applied LPNN to Robust TDOA source localization problem and Elliptic target positioning problem. Based on above mentioned researches, we try to solve problem (3) by using LPNN. The contributions are as follows:

- The nuclear norm minimization problem (NNM) is reformulated as a optimization problem of the matrix form, and we solve NNM by using a Lagrange programming neural network method (NNM-LPNN).

- By constructing an appropriate Lyapunov function, we propose the convergence conditions to prove the stability of NNM-LPNN.

## 2 Preliminaries

### 2.1 Notations

Suppose that $\mathbb{R}^{m \times n}$ is $m \times n$-dimensional real matrices. The capital $A$ is a matrix and $a_{(i,j)}$ is the the $i, j$-th entry of $A$. $A^\top$, $rank(A)$ and $Tr(A)$ are the transposition, the rank and the trace of $A$, respectively. $\|A\|_F$ and $\|A\|_*$ represent the Frobenius norm and the nuclear norm of $A$. $\sqrt{Tr(A^\top A)} = \sqrt{\sum_{i,j} a_{ij}^2}$, and $\|A\|_{\rho_{max}}$ is the largest singular value of $A$. $\langle A, B \rangle_F$ and $A \odot B$ are the Frobenius inner product and the Hadamard product of $A$ and $B$.

## 2.2 Subdifferential

The definition of subdifferential is put forward in [23].

**Definition 1**. *Let the function $f : \Omega^{m \times n} \to \mathbb{R}$ be convex and $\Omega$ be convex. For any $\Delta \in \mathbb{R}^{m \times n}$, we have*

$$f(Y) - f(X) \geq \langle Y - X, \Delta \rangle_F \quad \forall X, Y \in \Omega,$$

*so that $\Delta \in \partial f(X)$, where $\partial f(X)$ serves as the subdifferential of $f(X)$. The partial differential function of $f(X, Y)$ with regard to $X$ is denoted by the symbol $\nabla_X f(X, Y)$.*

**Lemma 1**. *Let $f(X) = \|X\|_*, \forall X, Y \in \mathbb{R}^{m \times n}$, we have*

$$\langle X - Y, \partial \|X\|_* - \partial \|Y\|_* \rangle_F \geq 0 \tag{4}$$

*Proof.* Let $X, Y \in R^{m \times n}$, according to Definition 1, we have

$$\langle Y - X, \partial \|X\|_* \rangle_F \leq \|Y\|_* - \|X\|_* \tag{5a}$$

$$\langle X - Y, \partial \|Y\|_* \rangle_F \leq \|X\|_* - \|Y\|_*. \tag{5b}$$

Add (5a) and (5b), we obtain

$$\langle X - Y, \partial \|X\|_* - \partial \|Y\|_* \rangle_F \geq 0. \tag{6}$$

A matrix $X$ can be precisely represented by its singular value decomposition (SVD)

$$X = \sum_{k=1}^{r} \rho_k u_k v_k^*, \tag{7}$$

where the elements $\rho_1, \rho_2, \ldots, \rho_r \geq 0$ are the singular values, the singular vectors are $u_1, u_2, \ldots, u_r \in R^m$ and $v_1, v_2, \ldots, v_r \in R^n$.

In [24], $Y$ has the form

$$Y = E + W, \tag{8}$$

where $W \in \{Z | P_U Z = 0, Z P_V = 0, \|Z\|_F \leq 1\}$. Obviously, $Y$ is a subgradient of $\|X\|_*$.

In this article, the subgradient of the nuclear norm is represented as follows:

$$Y = E. \tag{9}$$

**Remark 1**. *The computation of the subgradient of the nuclear norm can be derived using the formula in (8). In [24], minimum-norm subgradient can be used to ensure convergence. Since E is SVD decomposition from X, it is only necessary to minimize Y by minimizing $\|W\|_F$. Due to the positive definiteness of Frobenius norm, we can easily get $W = 0$.*

## 2.3 Lagrange programming neural network

Problems involving general nonlinear programming with equality requirements can be solved using the following LPNN method

$$\min_X f(X), \quad s.t. \quad h(X) = \mathbf{0} \tag{10}$$

where $f(X) : \mathbb{R}^{m \times n} \to \mathbb{R}$ is the objective function, $h(X) : \mathbb{R}^{m \times n} \to \mathbb{R}^{m \times n}$ describes the $m \times n$ constraint on equality. Suppose that both functions $f$ and $h$ are quadratically differentiable. A

Lagrangian function of problem (10) is constructed as follows:

$$\mathcal{L}_{ep} = f(X) + \langle \Lambda, h(X) \rangle_F, \tag{11}$$

$\Lambda$ is a matrix form of Lagrange multipliers. In the view of the structure of neural networks, variable and Lagrange neurons cannot belong to the similar types of neurons. In other words, the entries of $X$ and $\Lambda$ are stored in variable neurons and Lagrange neurons, respectively. The dynamics of neurons determined by

$$\tau_0 \frac{dX}{dt} = -\frac{d\mathcal{L}_{ep}}{dX}$$
$$\tau_0 \frac{d\Lambda}{dt} = \frac{d\mathcal{L}_{ep}}{d\Lambda}, \tag{12}$$

where $\tau_0$ is the circuit's time constant.

## 3 Lagrange programming neural network for the nuclear norm optimization

### 3.1 NNM properties

NNM can be formulated as

$$\begin{aligned} \min \quad & \|X\|_* \\ s.t. \quad & W \odot X = W \odot M. \end{aligned} \tag{13}$$

According to problem (13), we have the following proposition.

**Proposition 1**. *Let $X^\star$ an optimal solution of problem* (13) *if and only if there exists a $\Lambda^\star$ and the following conditions*

$$-W \odot \Lambda^\star \in \partial \|X\|_* \tag{14a}$$

$$0 = W \odot (X^\star - M) \tag{14b}$$

*satisfied.*

The conditions for the Karush-Kuhn-Tucker (KKT) equation are outlined in Proposition 1 by (14) in its entirety. The KKT requirements are both necessary and sufficient because the problem is convex.

### 3.2 NNM-LPNN dynamics

To avoid stability issues around equilibrium, we provide an enhanced term $\frac{1}{2}\|W \odot (X - M)\|_F^2$ in the Lagrange function as follows:

$$\mathcal{L}_{NNM} = \|X\|_* + \langle \Lambda, W \odot (X - M) \rangle_F + \frac{1}{2}\|W \odot (X - M)\|_F^2. \tag{15}$$

The objective value at an equilibrium $X^*$ is unaffected by the addition of the augmented term, because $W \odot (X^* - M) = \mathbf{0}$. The gradients of $\mathcal{L}_{nnm}$ are given by

$$\partial_X \mathcal{L}_{NNM} = Y + W \odot \Lambda + W \odot (X - M) \tag{16a}$$

$$\partial_\Lambda \mathcal{L}_{NNM} = W \odot (X - M) \tag{16b}$$

where $Y \in \partial \|X\|_*$.

A neurodynamic centralized method for solving(13) is

$$
\begin{cases}
\dot{X} & = -Y - W \odot \Lambda - W \odot (X - M) \\
\dot{\Lambda} & = W \odot (X - M).
\end{cases}
$$

(17)

**Theorem 1**. *Suppose that* $\{X^\star, \Lambda^*\}$ *serves as the LPNN dynamics of the equilibrium point of system* (17). *The KKT criterion of problem* (15) *is satisfied at the equilibrium point. Thus, the KKT criterion of NNM is necessary and sufficient. The stabilization point of* (17) *is consistent with the desired solution of problem* (15).

*Proof.* By the definition of stabilization points

$$
\frac{dX^\star}{dt} = \mathbf{0}, \frac{d\Lambda^\star}{dt} = \mathbf{0}.
$$

(18)

Based on (17) and (18), we can conclude that

$$
-Y^\star - W \odot \Lambda^\star - W \odot (X^\star - M) = \mathbf{0},
$$

(19a)

$$
W \odot (X^\star - M) = \mathbf{0}.
$$

(19b)

Then, we have

$$
-Y^\star - W \odot \Lambda^\star = \mathbf{0}.
$$

(20)

With $Y^\star \in \partial \| X^\star \|_*$, we have

$$
-W \odot \Lambda^\star \in \partial \| X^\star \|_*.
$$

(21)

Therefore, (16a) is also satisfied. Similarly, we can prove that (14b) leads to (19b). The proof is finished.

**Theorem 2**. *The neurodynamic method* (17) *globally converges to the optimal solution of the problem* (13).

*Proof.* Construct a Lyapunov function:

$$
V_1(X, \Lambda) = \frac{1}{2} \| W \odot (X - X^\star) \|_F^2 + \frac{1}{2} \| W \odot (\Lambda - \Lambda^\star) \|_F^2
$$

(22)

By definition, the above function $V_1(X, \Lambda)$ is semi-positive definite and it is radially unbounded in the domain of definition. The derivative of $V_1$ can be expressed as

$$
\dot{V}_1(X, \Lambda) = \langle W \odot (X - X^\star), W \odot \dot{X} \rangle_F + \langle W \odot (\Lambda - \Lambda^\star), W \odot \dot{\Lambda} \rangle_F
$$

(23)

From (17) and (19), we have

$$
\begin{aligned}
\dot{X} & = -(Y - Y^\star) - W \odot (\Lambda - \Lambda^\star) - W \odot (X - X^\star) \\
\dot{\Lambda} & = W \odot (X - X^\star).
\end{aligned}
$$

(24)

Substituting (24) into (23), we obtain

$$
\begin{aligned}
\dot{V}_1(X, \Lambda) \quad &= \langle W \odot (X - X^\star), -W \odot (Y - Y^\star) \\
&\quad - W \odot (\Lambda - \Lambda^\star) - W \odot (X - X^\star) \rangle_F \\
&\quad + \langle W \odot (\Lambda - \Lambda^\star), W \odot (X - X^\star) \rangle_F \\
&= \langle W \odot (X - X^\star), -W \odot (Y - Y^\star) \\
&\quad - W \odot (X - X^\star) \rangle_F \\
&= -\langle W \odot (X - X^\star), W \odot (Y - Y^\star) \rangle_F \\
&\quad - \langle W \odot (X - X^\star), W \odot (X - X^\star) \rangle_F \\
&= -\langle W \odot (X - X^\star), W \odot (Y - Y^\star) \rangle_F - \|\dot{\Lambda}\|_F^2.
\end{aligned}
\tag{25}
$$

From Lemma 1 and the definition of $W$, we could get

$$
\langle W \odot (X - X^\star), W \odot (Y - Y^\star) \rangle_F \geq 0.
\tag{26}
$$

According to (26) and $\|\dot{\Lambda}\|_F^2 \geq 0$, we have $\dot{V}_1(X, \Lambda) \leq 0$. In other word, for arbitrary initial value $\{X, \Lambda\} \in \mathbb{R}^{m \times n}$, $V_1(X, \Lambda)$ is nonincreasing when $t \to +\infty$. Hence, the proof is completed.

## 4 Experiments

We carry out experimental simulations of the following aspects to verify the performance of the proposed approach (NNM-LPNN in short):

1. Optimality and convergence.

2. Mean Square Error (MSE), Normalized Mean Square Error (NMSE), Recovery Error (RE) and Peak signal-to-noise ratio (PSNR).

3. Recovery ability of low-rank images at different sampling rates (SRs) and comparison with classical Singular Value Thresholding (SVT) approach and alternating direction multiplier method (ADMM) [25] and fixed-point continuation(FPC) [9].

### 4.1 Low-rank numerical matrix recovery

*Example* 1: $M \in \mathbb{R}^{m \times n}$ is synthesized obeying distributed standard Gaussian distribution [4]. Giving a rank $r$, we generate two matrices $M_{m \times r}$ and $M_{r \times n}$ using NNM-LPNN. Suppose that $m = n = 50$ and $r = 5$. The matrix reconstruction result is shown in Fig 1. The recovery results with SR = 50% are shown. This phenomenon indicates that NNM-LPNN can recover the target $M$ accurately. Moreover, the optimal solution of problem (13) is shown in Fig 2. Its reconstruction error and relative error are shown in Figs 3 and 4. In summary, the NNM-LPNN can reach the equilibrium point and achieve the optimal solution.

### 4.2 Application of image reconstruction

*Example* 2: We select a real image as shown in Fig 5 and its first 50 singular values are shown in Fig 5. This image can be considered to be a low-rank image. As shown in Fig 7, we sampled the image at SR = {10%, 50%, 90%} and the reconstruction results of the three approaches are

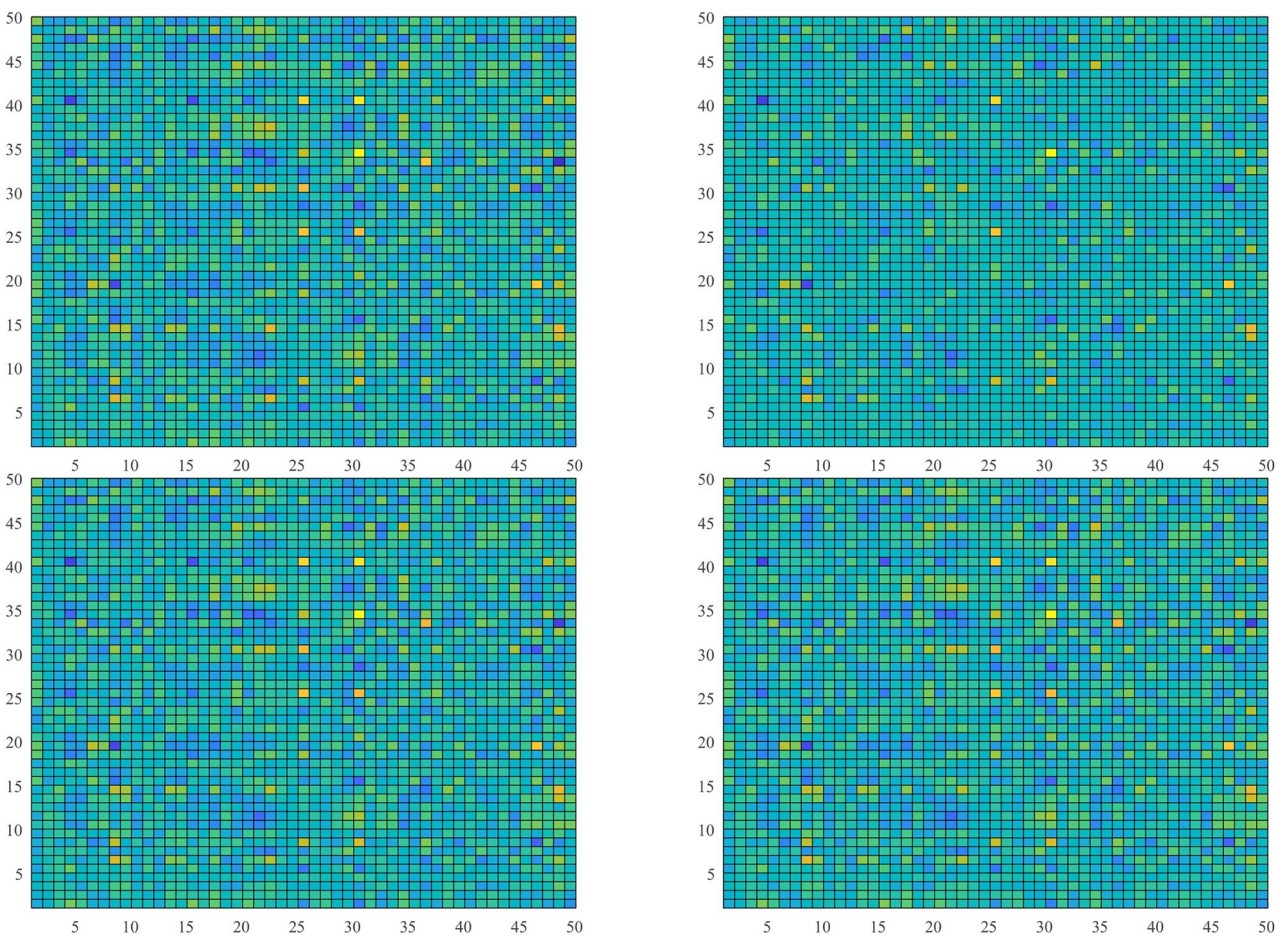

**Fig 1. Low-rank matrix recovery using NNM-LPNN after sampling (SR = 50%).**

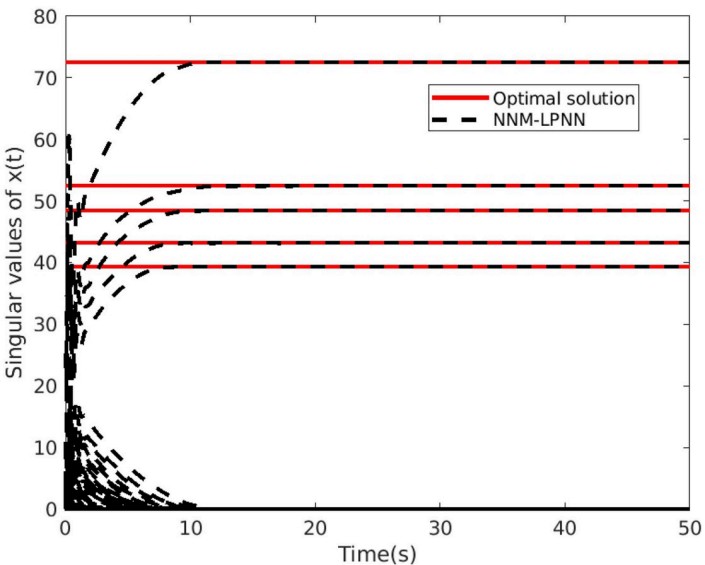

**Fig 2. Convergence to the optimal solution of NNM-LPNN.**

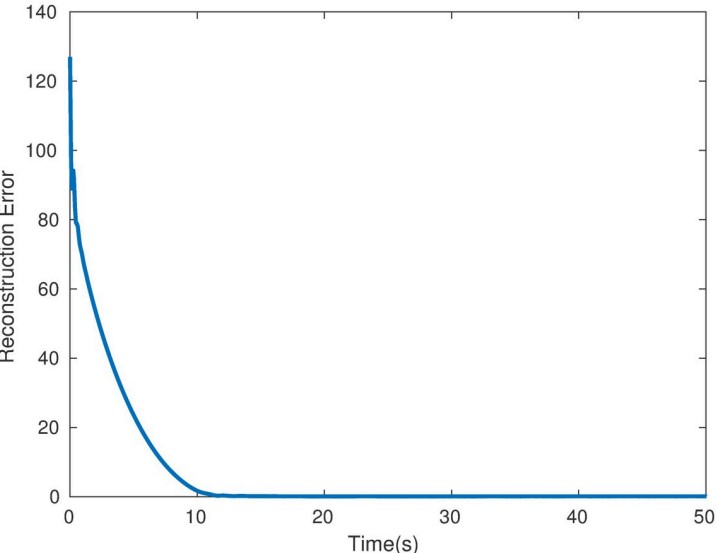

**Fig 3. The converge time to the equilibrium point of NNM-LPNN.**

shown in Fig 7. In addition, we have compared the Mean Square Error (MSE), the Normalised Mean Square Error (NMSE) and the Peak Signal-to-Noise Ratio (PSNR) of the proposed methods with the classical approach SVT, where $MSE = \frac{\|X-M\|_F^2}{m \times n}$, $NMSE = \frac{\|X-M\|_F^2}{\|X\|_F^2}$ and $PSNR = 10 \times \log\left(\frac{255^2}{MSE}\right)$.

Process the original image Fig 6. As shown in the Fig 7, NNM-LPNN can reconstruct the approximate low-rank real image. Since the reconstruction effects of the four approaches are similar visually, we analyze the reconstruction from the indicators data. For MSE and NMSE, the smaller the value is, the better. PSNR is opposite. In Tables 1–4, we can find that the effect

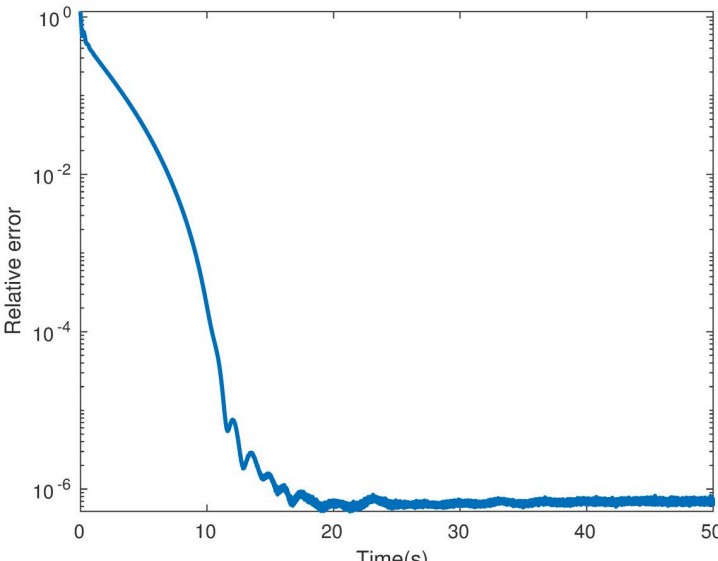

**Fig 4. Relative error between the recovered matrix and the low-rank matrix.**

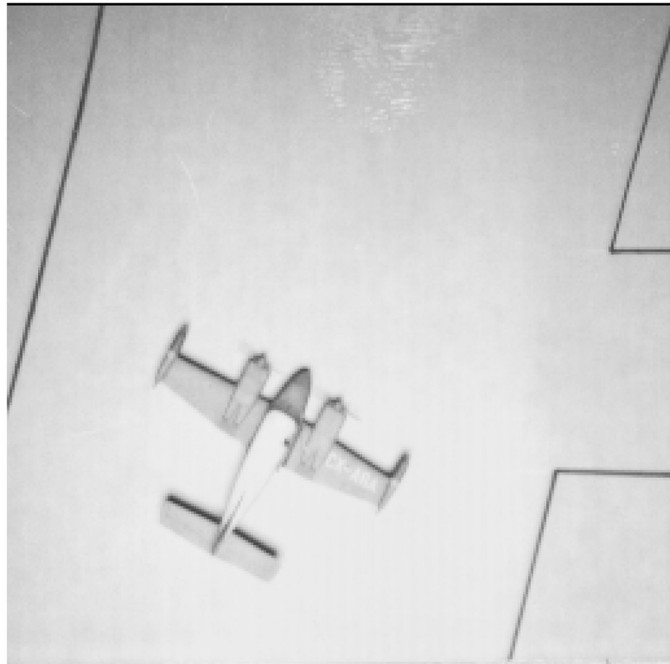
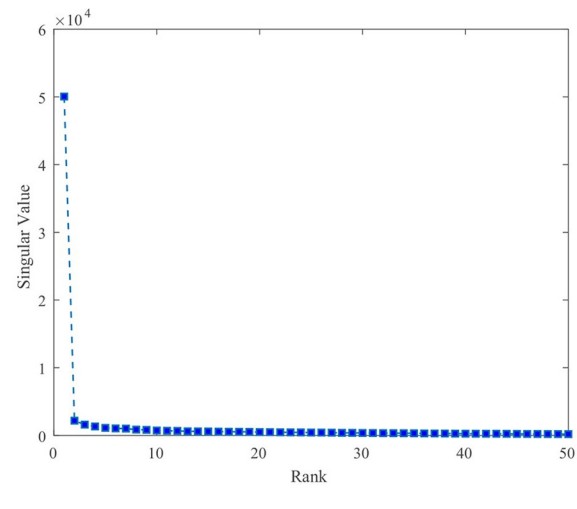

**Fig 5. The first 50 singular values of the Airplane.**

of NNM-LPNN is always better than the other two approaches, and occasionally FPC can be as good as NNM-LPNN. In a word, the NNM-LPNN proposed in this paper are better than the traditional SVT approach in reconstructing real images. Their results are shown in Figs 8–10.

## 5 Conclusion

In this article, a Lagrange programming neural network was proposed to solve the nuclear norm minimization. The stability and optimality of the proposed approaches were proved theoretically. Low-rank matrix recovery and approximate low-rank image recovery experiments demonstrated that the presented neural network is effective.

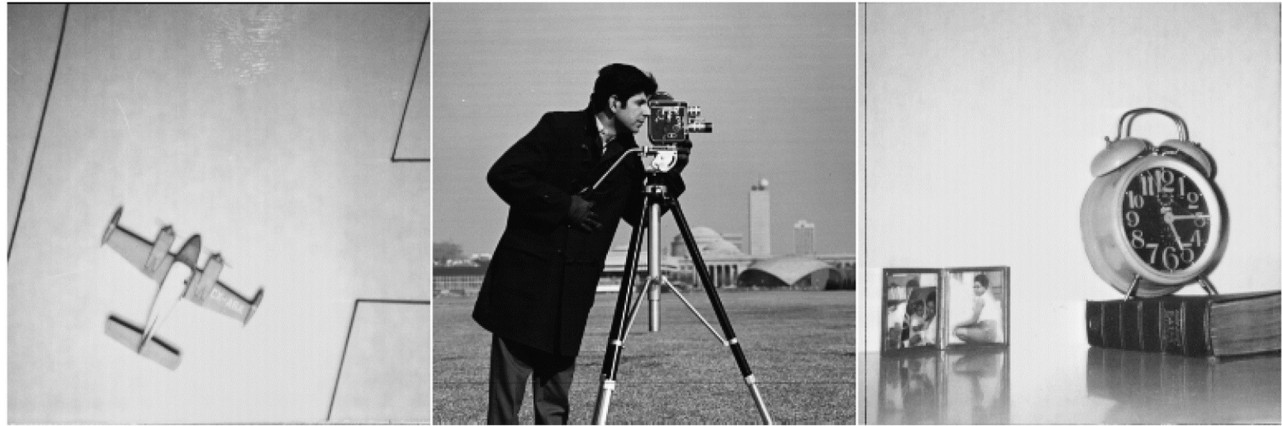

**Fig 6. Original image.**

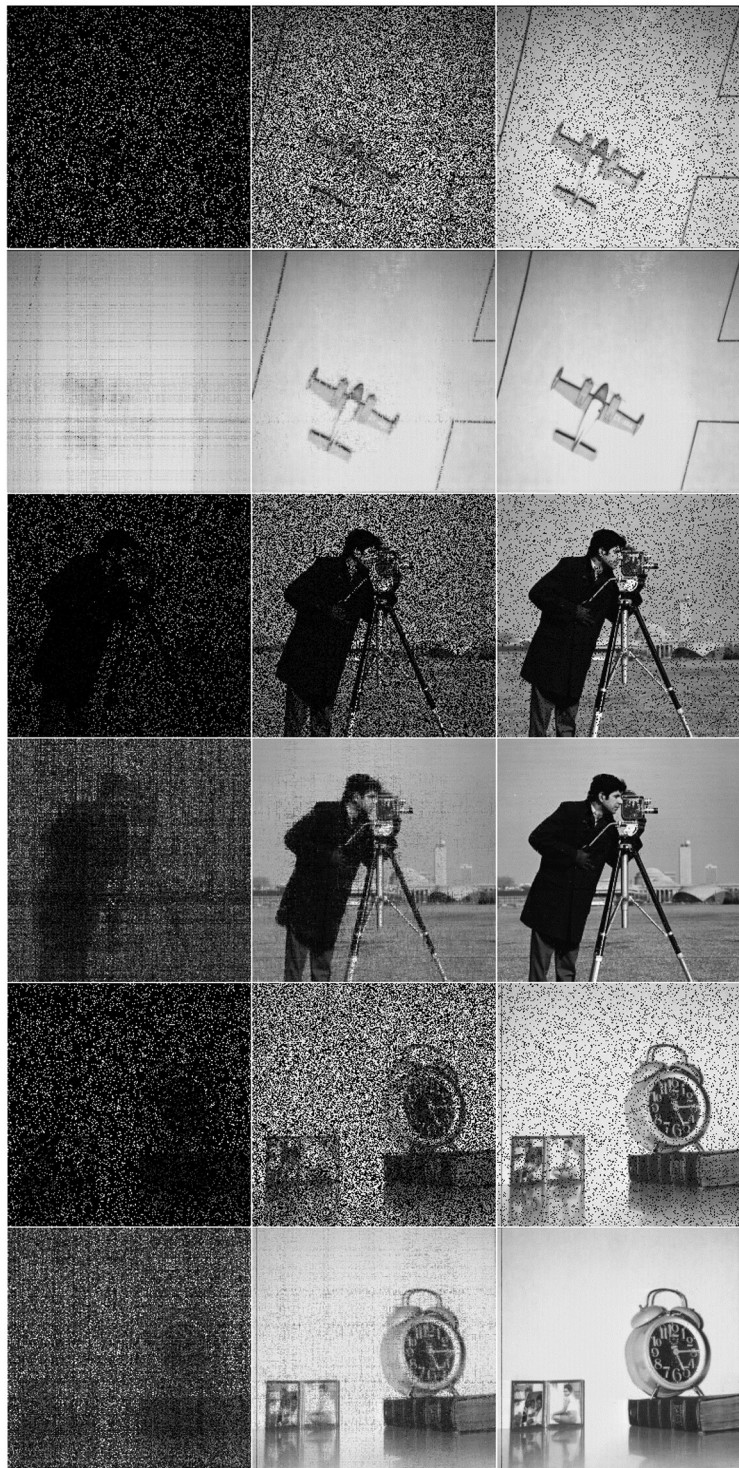

**Fig 7. Low-rank image reconstruction results under different sampling information with SR = {10%, 50%, 90%}.**

**Table 1. MSE, NMSE and PSNR of LPNN with SR = [10:10:90](%).**

| Indicators | MSE | NMSE | PSNR |
|---|---|---|---|
| SR = 10% | 386.58 | 1.00E-02 | 21.71 |
| SR = 20% | 218.66 | 5.67E-03 | 24.27 |
| SR = 30% | 139.26 | 3.61E-03 | 26.28 |
| SR = 40% | 84.30 | 2.19E-03 | 28.70 |
| SR = 50% | 51.27 | 1.33E-03 | 30.74 |
| SR = 60% | 32.55 | 8.44E-04 | 32.45 |
| SR = 70% | 19.64 | 5.09E-04 | 34.62 |
| SR = 80% | 8.64 | 2.24E-04 | 38.14 |
| SR = 90% | 2.91 | 7.54E-05 | 42.94 |

**Table 2. MSE, NMSE and PSNR of SVT with SR = [10:10:90](%).**

| Indicators | MSE | NMSE | PSNR |
|---|---|---|---|
| SR = 10% | 660.23 | 1.71E-02 | 19.43 |
| SR = 20% | 294.87 | 7.65E-03 | 23.03 |
| SR = 30% | 167.26 | 4.34E-03 | 25.43 |
| SR = 40% | 99.48 | 2.58E-03 | 27.70 |
| SR = 50% | 66.40 | 1.64E-03 | 29.65 |
| SR = 60% | 36.95 | 9.58E-04 | 31.94 |
| SR = 70% | 19.61 | 5.09E-04 | 34.65 |
| SR = 80% | 11.50 | 2.98E-04 | 36.84 |
| SR = 90% | 5.44 | 1.41E-04 | 40.07 |

**Table 3. MSE, NMSE and PSNR of ADMM with SR = [10:10:90](%).**

| Indicators | MSE | NMSE | PSNR |
|---|---|---|---|
| SR = 10% | 383.57 | 1.00E-02 | 21.25 |
| SR = 20% | 219.34 | 5.75E-03 | 24.31 |
| SR = 30% | 142.85 | 3.69E-03 | 26.24 |
| SR = 40% | 85.61 | 2.22E-03 | 28.56 |
| SR = 50% | 55.97 | 1.38E-03 | 30.15 |
| SR = 60% | 34.58 | 8.79E-04 | 32.45 |
| SR = 70% | 18.31 | 5.19E-04 | 35.94 |
| SR = 80% | 9.88 | 2.46E-04 | 37.64 |
| SR = 90% | 4.32 | 9.02E-05 | 41.93 |

**Table 4. MSE, NMSE and PSNR of FPC with SR = [10:10:90](%).**

| Indicators | MSE | NMSE | PSNR |
|---|---|---|---|
| SR = 10% | 364.04 | 1.00E-02 | 21.89 |
| SR = 20% | 212.38 | 5.70E-03 | 24.47 |
| SR = 30% | 139.59 | 3.70E-03 | 26.46 |
| SR = 40% | 82.28 | 2.16E-03 | 28.81 |
| SR = 50% | 52.46 | 1.37E-03 | 30.48 |
| SR = 60% | 30.12 | 7.85E-04 | 32.88 |
| SR = 70% | 18.81 | 4.89E-04 | 34.90 |
| SR = 80% | 8.86 | 2.30E-04 | 38.24 |
| SR = 90% | 3.45 | 8.96E-05 | 41.99 |

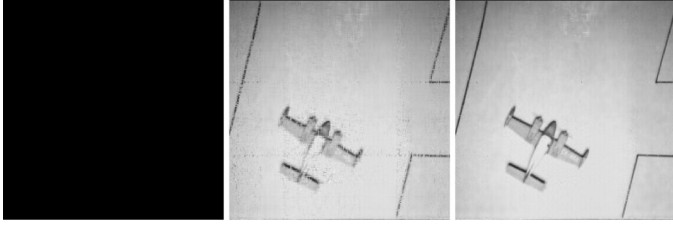

**Fig 8. Low-rank image reconstruction results under SVT.**

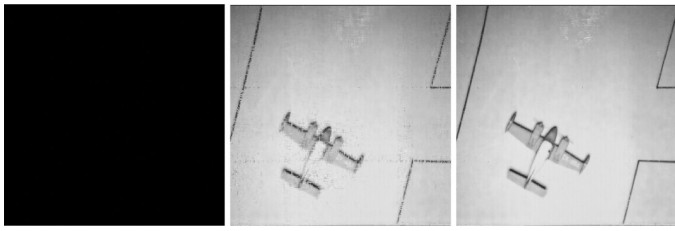

**Fig 9. Low-rank image reconstruction results under ADMM.**

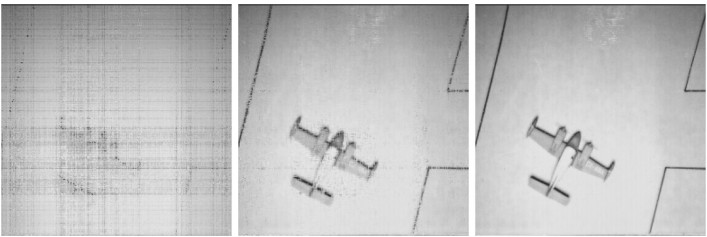

**Fig 10. Low-rank image reconstruction results under FPC.**

## Author Contributions

**Conceptualization:** Xiangguang Dai, Jian Qiu, Chaoyang Wan.

**Data curation:** Facheng Dai.

**Formal analysis:** Xiangguang Dai, Facheng Dai.

**Funding acquisition:** Facheng Dai.

**Investigation:** Facheng Dai.

**Methodology:** Facheng Dai.

**Project administration:** Facheng Dai.

**Software:** Xiangguang Dai.

**Supervision:** Facheng Dai.

**Validation:** Facheng Dai.

**Visualization:** Facheng Dai.

**Writing – original draft:** Jian Qiu, Chaoyang Wan, Facheng Dai.

**Writing – review & editing:** Xiangguang Dai, Jian Qiu, Chaoyang Wan.

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
