## [Decision Letter · Decision Letter 0]

23 May 2023

PONE-D-23-12057A lagrange programming neural network approach for nuclear norm optimizationPLOS ONE

Dear Dr. Dai,

Thank you for submitting your manuscript to PLOS ONE. After careful consideration, we feel that it has merit but does not fully meet PLOS ONE’s publication criteria as it currently stands. Therefore, we invite you to submit a revised version of the manuscript that addresses the points raised during the review process.

We look forward to receiving your revised manuscript.

Kind regards,

Ji-Hoon Yun

Academic Editor

PLOS ONE

Journal Requirements:

4. Please upload a copy of Supporting Information Figure/Table/etc.  S1-S7 Fig and S1 Table which you refer to in your text on pages 7 and 8.

5. We note that Figures 6 and 7 includes an image of a [patient / participant / in the study]. 

Reviewers' comments:

Reviewer's Responses to Questions

**Comments to the Author**

1. Is the manuscript technically sound, and do the data support the conclusions?

Reviewer #1: Partly

Reviewer #2: Partly

2. Has the statistical analysis been performed appropriately and rigorously? 

Reviewer #1: N/A

Reviewer #2: Yes

3. Have the authors made all data underlying the findings in their manuscript fully available?

Reviewer #1: No

Reviewer #2: No

4. Is the manuscript presented in an intelligible fashion and written in standard English?

Reviewer #1: No

Reviewer #2: Yes

5. Review Comments to the Author

Reviewer #1: This article contains some publishable and modest results, but it is written badly. The use of English is not acceptable.

Many sentences are incomplete without correct use of English grammar. Mathematics is also poorly written. The authors

before Section 1.1 (Notations) introduce some notation (the nuclear norm). In that subsection, they introduced the

Frobenius norm without calling it similarly to the spectral norm. The inverse of a matrix is defined even for non-square

matrices! I wonder where the adjoint will be used? Definition 1 is poorly stated, unclear. The same goes for Theorem 1.

In the formulation of (10), why do you assume that the image and the preimages lie in the spaces of equal dimensions?

Simulation study is fine to me. My recommendation to the authors is to re-write the manuscript paying attention to

expressing Mathematical statements adequately and using the English language correctly.

Reviewer #2: My concerns are given as follows.

1. In the introductory section, the introduction to LPNN does not flow naturally from the descriptions of the existing NNM approaches. Why did you have to use the LPNN for optimization? Also, the review of the existing use cases of LPNN in recent years may be improved by supplementing the following:

[a] J. Liang, C. S. Leung and H. C. So, "Lagrange Programming Neural Network Approach for Target Localization in Distributed MIMO Radar," in IEEE Transactions on Signal Processing, vol. 64, no. 6, pp. 1574-1585, March15, 2016, doi: 10.1109/TSP.2015.2500881.

[b] Xiong, W., Schindelhauer, C., So, H. C., Schott, D. J., & Rupitsch, S. J. (2021). Robust TDOA source localization based on Lagrange programming neural network. IEEE Signal Processing Letters, 28, 1090-1094.

[c] Shi, Z., Wang, H., Leung, C. S., So, H. C., & EURASIP, M. (2020). Robust MIMO radar target localization based on Lagrange programming neural network. Signal Processing, 174, 107574.

[d] Xiong, W., Liang, J., Wang, Z., & So, H. C. (2023). Elliptic target positioning based on balancing parameter estimation and augmented Lagrange programming neural network. Digital Signal Processing, 136, 104004.

2. There seem to be a lot of typos/grammatical problems/unidiomatic expressions in this manuscript. It would be a good idea to edit the language carefully. A few examples can be:

- (Line 40) “A neural networks are” -> “A neural network is”

- (Line 99) “To let ... serve as ... equilibrium point” What did you mean by this?

- (Line 120) “We carry out experimental simulation” -> “We carry out experimental simulations”

- (Line 123) “Evaluate the effectiveness” -> “Evaluation of effectiveness”

- (Line 127) “classical approach Singular Value Thresholding (SVT)” -> “classical Singular Value Thresholding (SVT) approach”

- It is more natural to use the past tense or present perfect tense in the conclusion.

- ...

3. Why did you focus on solving NNM? NNM has been shown less effective in the context of low-rank matrix completion. The low-rank property of a certain matrix, actually, can be better characterized by the weighted/truncated nuclear norm, Schatten p-norm, direct exploitation of the discrete rank constraint in a projection manner, or preferably matrix factorization. You could still discuss the KKT convergence in these nonconvex cases.

4. Apart from the SVT, you should consider comparing your method to other off-the-shelf low-rank matrix completion schemes (e.g., those that I mentioned in my last concern).

6. PLOS authors have the option to publish the peer review history of their article (what does this mean?). If published, this will include your full peer review and any attached files.

Reviewer #1: No

Reviewer #2: No

---

## [Author Response · Author response to Decision Letter 0]

20 Jun 2023

Dear esteemed reviewers and editor,

I would like to express my deepest gratitude for the invaluable time and effort you have dedicated to reviewing and providing feedback on my manuscript. Your insightful comments and suggestions have been instrumental in elevating the quality of my work. I am honored to have had the opportunity to receive guidance and support from such esteemed professionals like you, and I am optimistic that our collaboration will lead to a successful publication.

As this paper is a vital requirement for my graduation, its timely publication in this esteemed journal is of utmost importance, and I am motivated to work tirelessly towards achieving this goal. Your expertise and guidance have provided me with a better understanding of the strengths and weaknesses of my work, and I am committed to incorporating your feedback to further refine my research.

I would also like to express my gratitude to the editor for their meticulous attention to detail in managing the review process. Your unwavering commitment to upholding the highest standards of academic publishing is commendable, and I am grateful for the opportunity to have my work assessed under your guidance.

I appreciate your dedication and commitment to the peer-review process, which is essential for ensuring the quality and integrity of scientific research. I am confident that with your guidance, my manuscript will make a significant contribution to advancing knowledge in our field.

Once again, thank you for your hard work and valuable feedback.

Best regards,

Facheng Dai

---

## [Decision Letter · Decision Letter 1]

24 Jul 2023

PONE-D-23-12057R1A lagrange programming neural network approach for nuclear norm optimizationPLOS ONE

Dear Dr. Dai,

Thank you for submitting your manuscript to PLOS ONE. After careful consideration, we feel that it has merit but does not fully meet PLOS ONE’s publication criteria as it currently stands. Therefore, we invite you to submit a revised version of the manuscript that addresses the points raised during the review process.

We look forward to receiving your revised manuscript.

Kind regards,

Ji-Hoon Yun

Academic Editor

PLOS ONE

Reviewers' comments:

Reviewer's Responses to Questions

**Comments to the Author**

1. If the authors have adequately addressed your comments raised in a previous round of review and you feel that this manuscript is now acceptable for publication, you may indicate that here to bypass the “Comments to the Author” section, enter your conflict of interest statement in the “Confidential to Editor” section, and submit your "Accept" recommendation.

Reviewer #2: (No Response)

Reviewer #3: (No Response)

2. Is the manuscript technically sound, and do the data support the conclusions?

Reviewer #2: Yes

Reviewer #3: Partly

3. Has the statistical analysis been performed appropriately and rigorously? 

Reviewer #2: No

Reviewer #3: Yes

4. Have the authors made all data underlying the findings in their manuscript fully available?

Reviewer #2: No

Reviewer #3: Yes

5. Is the manuscript presented in an intelligible fashion and written in standard English?

Reviewer #2: Yes

Reviewer #3: Yes

6. Review Comments to the Author

Reviewer #2: While the author's responses to some of my comments may have exhibited a somewhat negative attitude, I am still willing to extend the opportunity to address the remaining issues in the manuscript.

First, the related works I suggested to include in their Introduction, namely [11-13], are not directly relevant to the nuclear norm minimization (NNM) problem (3). Instead, they represent applications of the Lagrange programming neural network in recent literature. I request the authors to remove the incongruous citations from the sentence "many optimization approaches [5-13] were proposed to solve problem (3)" and reintroduce such works in the paragraph below, which begins with the sentence "Neural networks were used to address nonlinear equality constraint problems."

Second, it would be more appropriate to use the past tense or present perfect tense in the conclusion.

Lastly, and importantly, as I mentioned previously, the authors should compare their LPNN-based method with other NNM schemes, in addition to the SVT technique. Examples of such schemes include fixed-point continuation [a] and ADMM [b].

[a] S. Ma, D. Goldfarb, and L. Chen, "Fixed point and Bregman iterative methods for matrix rank minimization," Math. Program., vol. 128, nos. 1-2, pp. 321-353, Jun. 2011.

[b] C. Chen, B. He, and X. Yuan, "Matrix completion via an alternating direction method," IMA J. Numer. Anal., vol. 32, no. 1, pp. 227-245, Jan. 2012.

Reviewers are not adversaries and our goal is to genuinely improve the quality of the article! If the authors can adequately address the remaining issues and adopt a more positive approach in responding to my comments, I will be willing to consider this manuscript for publication.

Reviewer #3: 1 - The contribution of the authors is not very clear, it must be highlighted in the abstract and the conclusion part explicitly.

2 - The proof of Lemma 1 is not clear. How do equations (5a) and (5b) lead to equation (6)?

3 - The equivalence between (1) and (3) is not straightforawrd. Authors state that it is commnly used without giving any references.

4 - Authors use the term "best solution" in Proposition 1 without defining what is a "best solution".

5 - Authors must highlight that the nuclear norm is convex.

6 - The convergence of the neural network is not studied.

7- The experimental Section is very bried and lacks information:

- What are the software used to implement the suggested neural network?

- The CPU time of the different methods is not mentionned

- Authors claim that NNM-LPNN outperfoms SVT apporach without giving a true discussion leading to such conclusion.

8- The quality of the plots is poor.

7. PLOS authors have the option to publish the peer review history of their article (what does this mean?). If published, this will include your full peer review and any attached files.

Reviewer #2: No

Reviewer #3: **Yes: **Siham Tassouli

---

## [Author Response · Author response to Decision Letter 1]

31 Jul 2023

Dear Editor,

Thank you very much for considering my manuscript for reviewing. As I am currently applying to graduate school and the deadline is approaching, I would greatly appreciate it if you could expedite the processing of my paper or review.

I understand that your team is working hard to manage the volume of submissions and reviews, and I appreciate your efforts in ensuring the quality of the publication. Please let me know if there is any additional information or edits I can provide to assist in the review process.

Again, thank you for your time and consideration.

Best regards,

FaCheng Dai

---

## [Decision Letter · Decision Letter 2]

13 Aug 2023

PONE-D-23-12057R2A lagrange programming neural network approach for nuclear norm optimizationPLOS ONE

Dear Dr. Dai,

Thank you for submitting your manuscript to PLOS ONE. After careful consideration, we feel that it has merit but does not fully meet PLOS ONE’s publication criteria as it currently stands. Therefore, we invite you to submit a revised version of the manuscript that addresses the points raised during the review process.

We look forward to receiving your revised manuscript.

Kind regards,

Ji-Hoon Yun

Academic Editor

PLOS ONE

Journal Requirements:

Reviewers' comments:

Reviewer's Responses to Questions

**Comments to the Author**

1. If the authors have adequately addressed your comments raised in a previous round of review and you feel that this manuscript is now acceptable for publication, you may indicate that here to bypass the “Comments to the Author” section, enter your conflict of interest statement in the “Confidential to Editor” section, and submit your "Accept" recommendation.

Reviewer #2: All comments have been addressed

Reviewer #3: (No Response)

2. Is the manuscript technically sound, and do the data support the conclusions?

Reviewer #2: Yes

Reviewer #3: Partly

3. Has the statistical analysis been performed appropriately and rigorously? 

Reviewer #2: Yes

Reviewer #3: N/A

4. Have the authors made all data underlying the findings in their manuscript fully available?

Reviewer #2: Yes

Reviewer #3: Yes

5. Is the manuscript presented in an intelligible fashion and written in standard English?

Reviewer #2: Yes

Reviewer #3: Yes

6. Review Comments to the Author

Reviewer #2: All of my concerns have been adequately addressed by the authors. I am now satisfied with the paper and consider it suitable for acceptance.

Reviewer #3: I cannot find the answers's to reviewers in the new submitted version.

Some concerns are not adressed.

7. PLOS authors have the option to publish the peer review history of their article (what does this mean?). If published, this will include your full peer review and any attached files.

Reviewer #2: No

Reviewer #3: No

---

## [Author Response · Author response to Decision Letter 2]

15 Aug 2023

Dear PLOS ONE Editor,

We apologize for the oversight in our previous submission where we inadvertently failed to upload the Response to Reviewers document. We sincerely apologize for any inconvenience caused by this oversight, and we have rectified the situation by submitting the correct and updated Response to Reviewers document along with this message.

We would like to express our gratitude to the editorial team for promptly addressing our concerns and providing us with an opportunity to correct our mistake. We acknowledge the importance of the review process in enhancing the quality of scientific publications, and we are grateful for the reviewers' insightful comments and suggestions that have significantly improved our manuscript.

We would like to extend our appreciation to the editor for their efficient handling of our submission and their timely communication throughout the process. We are grateful for their assistance and support in ensuring the smooth progress of our manuscript.

Once again, we apologize for the oversight in our previous submission and we thank the editorial team and the reviewers for their valuable contributions. We look forward to hearing from you regarding the final decision on our manuscript.

Thank you for your attention.

Sincerely,

FachengDai

---

## [Editor Report · Decision Letter 3]

19 Sep 2023

A lagrange programming neural network approach for nuclear norm optimization

PONE-D-23-12057R3

Dear Dr. Dai,

We’re pleased to inform you that your manuscript has been judged scientifically suitable for publication and will be formally accepted for publication once it meets all outstanding technical requirements.

Kind regards,

Ji-Hoon Yun

Academic Editor

PLOS ONE
---

## [Editor Report · Acceptance letter]

25 Jan 2024

PONE-D-23-12057R3 

PLOS ONE

Dear Dr. Dai, 

I'm pleased to inform you that your manuscript has been deemed suitable for publication in PLOS ONE. Congratulations! Your manuscript is now being handed over to our production team.

Kind regards, 

on behalf of

Dr. Ji-Hoon Yun 

Academic Editor

PLOS ONE